

# Lattice simulations of the QCD chiral transition at real $\mu_B$

Attila Pásztor[1⋆], Szabolcs Borsányi[2], Zoltán Fodor[1,2,3,4,5], M. Giordano[1],
Sandor D. Katz[1,6], Daniel Nógrádi[1] and Chik Him Wong[2]

**1** ELTE Eötvös Loránd University, Institute for Theoretical Physics,
Pázmány Péter sétány 1/A, H-1117, Budapest, Hungary
**2** Department of Physics, Wuppertal University, Gaussstr. 20, D-42119, Wuppertal, Germany
**3** Pennsylvania State University, Department of Physics, State College,
Pennsylvania 16801, USA
**4** Jülich Supercomputing Centre, Forschungszentrum Jülich, D-52425 Jülich, Germany
**5** Physics Department, UCSD, San Diego, CA 92093, USA
**6** MTA-ELTE Theoretical Physics Research Group,
Pázmány Péter sétány 1/A, H-1117 Budapest, Hungary

⋆ apasztor@bodri.elte.hu

## Abstract

**Most lattice studies of hot and dense QCD matter rely on extrapolation from zero or imaginary chemical potentials. The ill-posedness of numerical analytic continuation puts severe limitations on the reliability of such methods. We studied the QCD chiral transition at finite real baryon density with the more direct sign reweighting approach. We simulate up to a baryochemical potential-temperature ratio of $\mu_B/T = 2.7$, covering the RHIC Beam Energy Scan range, and penetrating the region where methods based on analytic continuation are unpredictive. This opens up a new window to study QCD matter at finite $\mu_B$ from first principles. This conference contribution is based on Ref. [1].**

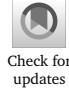

## 1 Introduction

### 1.1 QCD at finite $\mu_B$ and the need for more direct methods

One of the major unsolved problems in high energy physics is the calculation of the phase diagram of strongly interacting matter in the temperature ($T$) - baryochemical potential ($\mu_B$) plane. Many aspects of QCD thermodynamics at $\mu_B = 0$ have been clarified by first principle lattice QCD calculations, such as the crossover nature of the transition and the value of the transition temperature [2–4].

It is conjectured that at higher baryochemical potential the QCD crossover gets stronger and above a certain point turns into a first order phase transition. The endpoint of the line of first order transitions is called the critical endpoint. Establishing the existence and the location of this conjectured critical endpoint is one of the main goals of the phenomenology of heavy ion collisions and of QCD thermodynamics.

First principle lattice calculations at finite $\mu_B$ are, however, hampered by the notorious complex-action problem: the path integral weights become complex numbers, and importance sampling breaks down. A number of methods have been introduced over the years to side-step this problem. In particular, most state-of-the-art calculations involve analytic continuation using either i) data on Taylor coefficients of different observables at $\mu_B = 0$ or ii) data at purely imaginary chemical potentials $\mu_B^2 \leq 0$, where the sign problem is absent. An example of an important result coming from these approaches is the calculation of the curvature of the crossover line $T_c(\mu_B)$ near zero chemical potential [5–7]. Another important result is the calculation of the Taylor coefficients of the pressure in a series expansion in the chemical potential up to fourth order [8,9], which have been calculated on the lattice up to high enough temperatures to match results from resummed perturbation theory [10, 11].

The extension of these results to higher orders in the Taylor expansion and to higher chemical potentials, however, faces immense challenges: For the Taylor method, the signal-to-noise ratio increases significantly with increasing order of the Taylor expansions. Similarly, in the determination of the same high-order coefficients with the imaginary chemical potential method, one runs into the ill-posedness of high-order numerical differentiation. Even if the high-order coefficients were available, extrapolation by a Taylor polynomial ansatz is limited by the radius of convergence of such an expansion. While there were attempts to locate the leading singularity of the pressure with several different methods [12–15], these calculations have so far not reached the continuum limit. Even if one knows the leading singularity determining the radius of convergence, it is not obvious how to go beyond it. Several resummation schemes have been experimented with, including Padé resummation in Refs. [15–17], a joint expansion in temperature and chemical potential along lines of constant physics in Ref. [18], and a truncated reweighting scheme in Ref. [14]. While these methods are interesting, at the moment they provide no clear way of going beyond the crossover region of the conjectured phase diagram. Moreover, these type of reweighting schemes have so far been used mostly to calculate observables that are not very sensitive to criticality - such as the pressure and the transition line $T_c(\mu_B)$. Extrapolations of observables that are sensitive to criticality, such as the width of the transition, are even less under control [7].

To shed light on the ultimate fate of the QCD crossover at finite $\mu_B$, it is therefore of great importance to come up with more direct methods, that can provide results directly at a finite chemical potential, and are free of additional systematic effects, such as the aforementioned analytic continuation problem of the Taylor and imaginary chemical potential methods, or the convergence issues of complex Langevin [19–21].

## 1.2 Reweighting and the overlap problem

Given a theory with fields $U$, reweighting is a general strategy to calculate expectation values in a target theory - with path integral weights $w_t$ and partition function $Z_t = \int \mathcal{D}U w_t(U)$ - by performing simulations in a different (simulated) theory - with path integral weights $w_s$ and partition function $Z_s = \int \mathcal{D}U w_s(U)$. The ratio of the partition functions and expectation value in the target theory are then given by

$$\frac{Z_t}{Z_s} = \left\langle \frac{w_t}{w_s} \right\rangle_s \quad \text{and} \quad \langle \mathcal{O} \rangle_t = \frac{\left\langle \frac{w_t}{w_s} \mathcal{O} \right\rangle_s}{\left\langle \frac{w_t}{w_s} \right\rangle_s} \tag{1}$$

respectively, where $\langle\ldots\rangle_{t,s}$ denotes taking expectation value with respect to the weights $w_t$ and $w_s$, respectively. In the present conference contribution, we will consider examples where the target theory is QCD at finite baryochemical potential discretized on the lattice. In this case the partition function of the target theory is:

$$Z_\mu = \int \mathcal{D}U \det M(U,\mu,m)e^{-S_g(U)} = \int \mathcal{D}U \operatorname{Re} \det M(U,\mu,m)e^{-S_g(U)}, \tag{2}$$

where $S_g$ is the gauge action, $\det M$ denotes the fermionic determinant, including all quark types with their respective masses collectively denoted by $m$, their respective chemical potentials collectively denoted by $\mu$, as well as rooting in the case of staggered fermions, and the integral is over all link variables $U$. Replacing the determinant with its real part is not permitted for arbitrary expectation values, but it is allowed for i) observables satisfying either $\mathcal{O}(U^*) = \mathcal{O}(U)$ or ii) observables obtained as derivatives of $Z$ with respect to real parameters, such as the chemical potential, the quark mass or the gauge coupling.

Since the target theory is lattice QCD at finite chemical potential, the weights $w_t$ have wildly fluctuating phases: this is the infamous sign problem of lattice QCD at finite baryon density. In addition to this problem, generic reweighting methods also suffer from an overlap problem: the probability distribution of the reweighting factor $w_t/w_s$ has generally a long tail, which cannot be sampled efficiently in standard Monte Carlo simulations.

Many attempts at reweighting to finite baryochemical potential, such as Refs. [13, 22–24] use reweighting from zero chemical potential, when the weights are proportional to the ratio of determinants $\det M(\mu)/\det M(0)$. However, these studies have so far been restricted to coarse lattices, with temporal extent $N_\tau = 4$, and mostly an unimproved staggered action, with the exception of Ref. [13], that uses the 2stout improved staggered action [3], albeit still at $N_\tau = 4$. It was actually demonstrated in Ref. [25], that the main bottleneck in extending such studies to finer lattices is the overlap problem in the weights $w_t/w_s$, which becomes severe already at moderate chemical potentials, where the sign problem is still numerically manageable.

This overlap problem in the weights $w_t/w_s$ is not present if they take values in a compact space. The most well-known of these approaches is phase reweighting [26, 27], where the simulated theory - the so called phase quenched theory - has path integral weights:

$$w_s = w_{PQ} = |\det M_{ud}(\mu)^{\frac{1}{2}}|\det M_s(0)^{\frac{1}{4}}e^{-S_g}. \tag{3}$$

In this case the reweighting factors are pure phases:

$$\left(\frac{w_t}{w_s}\right)_{PQ} = e^{i\theta}, \tag{4}$$

where $\theta = \operatorname{Arg}\det M$. We will contrast this approach with sign reweighting, where the simulated - sign quenched - ensemble has weights:

$$w_s = w_{SQ} = |\operatorname{Re}\det M_{ud}(\mu)^{\frac{1}{2}}|\det M_s(0)^{\frac{1}{4}}e^{-S_g}. \tag{5}$$

In this case the reweighting factor are signs:

$$\left(\frac{w_t}{w_s}\right)_{SQ} = \epsilon \equiv \operatorname{sign}\cos\theta = \pm 1, \tag{6}$$

provided that the target theory is the one with $w_t = \operatorname{Re}\det M e^{-S_g}$, i.e., provided one restricts one's attention to observables satisfying i) or ii).

## 2  The severity of the sign problem

A measure of the strength of the sign problem in the phase reweighting scheme is given by the expectation value of the phases $\frac{Z_\mu}{Z_{PQ}} = \langle \cos\theta \rangle_{PQ}$. Similarly, in the sign reweighting scheme the severity of the sign problem is measured by $\frac{Z_\mu}{Z_{SQ}} = \langle \epsilon \rangle_{SQ}$. The earliest mention of the sign reweighting approach we are aware of is Ref. [28], where it was noted that out of the reweighting schemes where the weights $w_t/w_s$ are a function of the phase of the quark determinant only, sign reweighting is the optimal one, with the weakest sign problem, in the sense that the ratio $Z_t/Z_s$ is maximal. In this section we study how much one gains by this optimality property of the sign quenched ensemble, when compared to the phase quenched ensemble. For this purpose we introduce a simplified model - to be later compared with direct simulation data - where the distribution of the phases $\theta$ in the phase quenched ensemble is given by a wrapped Gaussian distribution:

$$P_{\text{PQ}}(\theta) \underset{\substack{\text{Gaussian}\\\text{approx.}}}{=} \frac{1}{\sqrt{2\pi}\sigma} \sum_{n=-\infty}^{\infty} e^{-\frac{1}{2\sigma^2}(\theta+2\pi n)^2}. \tag{7}$$

Once one has a model for this probability distribution, the strength of the sign problem can be estimated in both the phase and sign quenched ensembles. The estimates and their small chemical potential (i.e., small $\sigma$) asymptotics are given by:

$$
\begin{aligned}
\langle \cos\theta \rangle_{T,\mu}^{\text{PQ}} &= e^{-\frac{\sigma^2(\mu)}{2}} \underset{\mu_B \to 0}{\sim} 1 - \frac{\sigma^2(\mu)}{2}, \\
\langle \varepsilon \rangle_{T,\mu}^{\text{SQ}} &= \frac{\langle \cos\theta \rangle_{T,\mu}^{\text{PQ}}}{\langle |\cos\theta| \rangle_{T,\mu}^{\text{PQ}}} \underset{\mu_B \to 0}{\sim} 1 - \left(\frac{4}{\pi}\right)^{\frac{5}{2}} \left(\frac{\sigma^2(\mu)}{2}\right)^{\frac{3}{2}} e^{-\frac{\pi^2}{8\sigma^2(\mu)}}.
\end{aligned}
\tag{8}
$$

Note the two very different asymptotics at small chemical potential: the phase reweighting approach leads to a regular Taylor series, while in the sign reweighting approach the asymptotics approach 1 faster than any polynomial.

The large-$\mu$ or large volume asymptotics are on the other hand very similar: in the large-$\sigma$ limit a wrapped Gaussian tends to the uniform distribution, and so at large chemical potential or volume one arrives at

$$\frac{\langle \varepsilon \rangle_{T,\mu}^{\text{SQ}}}{\langle \cos\theta \rangle_{T,\mu}^{\text{PQ}}} \underset{\mu_B \text{ or } V \to \infty}{\sim} \left(\int_{-\pi}^{\pi} d\theta \, |\cos\theta|\right)^{-1} = \frac{\pi}{2}, \tag{9}$$

which asymptotically translates to a factor of $(\frac{\pi}{2})^2 \approx 2.5$ less statistics needed for a sign quenched as compared to a phase quenched simulation.

To have a numerical estimate of the strength of the sign problem as a function of $\mu$, rather than $\sigma$ we further approximate the variance of the weights by the leading order Taylor expansion [29]:

$$\sigma(\mu)^2 \approx \langle \theta^2 \rangle_{\text{LO}} = -\frac{4}{9} \chi_{11}^{ud} (LT)^3 \left(\frac{\mu_B}{T}\right)^2, \tag{10}$$

where

$$\chi_{11}^{ud} = \frac{1}{T^2} \frac{\partial^2 p}{\partial \mu_u \partial \mu_d}\Big|_{\mu_u = \mu_d = 0} \tag{11}$$

is the disconnected part of the light quark susceptibility, which is easily obtained by performing simulations at zero chemical potential.

The simple approximations made above are actually quite close to the actual simulation data, as can be seen in Fig. 1: our simple model predicts the strength of the sign problem both

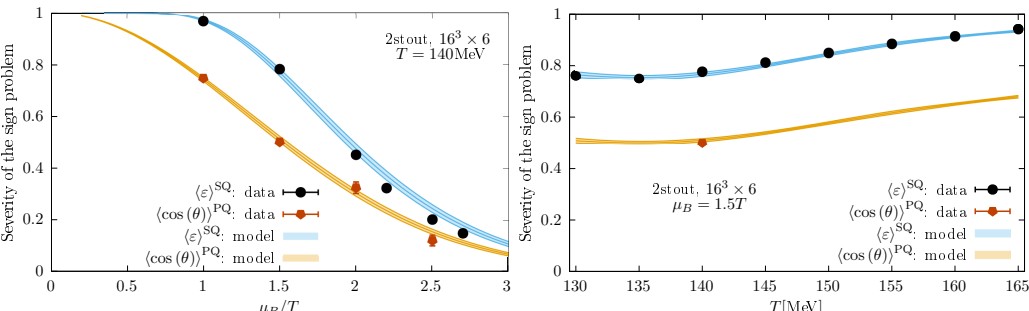

Figure 1: The strength of the sign problem on 2stout improved $16^3 \times 6$ staggered lattices as a function of $\mu_B/T$ at $T = 140$ MeV (left) and as a function of $T$ at $\mu_B/T = 1.5$. A value close to 1 shows a mild sign problem, while a small value indicates a severe sign problem. Data for sign reweighting (black) and phase reweighting (orange) are from simulations. Predictions of the Gaussian model (see text) are also shown.

as a function of $\mu_B$ at a fixed temperature (left) and as a function of temperature at a fixed $\mu_B/T$ (right). While deviations are visible at larger $\mu$, even at the upper end of our $\hat{\mu}_B \equiv \frac{\mu_B}{T}$ range the deviation is at most 25%, and Eq. (9) approximates well the relative severity of the sign problem in the two ensembles at $\mu_B/T > 1.5$. This is of great practical importance, as it makes the planning of future simulation projects with either the sign or phase reweighting approaches relatively straightforward: simulation costs can be easily estimated beforehand.

## 3 Lattice setup and numerical results

For the simulations we used a tree level Symanzik improved gauge action with the staggered Dirac operator being a function of fat links, obtained by two steps of stout smearing [30] with parameter $\rho = 0.15$. We only introduce a chemical potential for the up and down quarks, that have the same chemical potential $\mu = \mu_l = \mu_u = \mu_d = \mu_B/3$, while for the strange quark we have $\mu_s = 0$. We used a lattice size of $16^3 \times 6$, and performed a scan in chemical potential at fixed $T = 140$ MeV, and a scan in temperature at fixed $\mu_B/T = 1.5$. In both cases, simulations were performed by modifying the RHMC algorithm at $\mu_B = 0$ by including an extra accept/reject step that takes into account the factor $\frac{|\mathrm{Re}\det M_{ud}(\mu)^{\frac{1}{2}}|}{\det M_{ud}(0)}$. The determinant was calculated with the reduced matrix formalism [22] and dense linear algebra, with no stochastic estimators involved.

The main observables we studied were the light quark condensate and density. The light-quark chiral condensate was obtained via the formula

$$\langle \bar{\psi}\psi \rangle_{T,\mu} = \frac{1}{Z(T,\mu)} \frac{\partial Z(T,\mu)}{\partial m_{ud}} = \frac{T}{V} \frac{1}{\langle \varepsilon \rangle^{SQ}_{T,\mu}} \left\langle \varepsilon \frac{\partial}{\partial m_{\mathrm{ud}}} \ln \left| \mathrm{Re}\det M^{\frac{1}{2}}_{ud} \right| \right\rangle^{SQ}_{T,\mu}, \qquad (12)$$

using a numerical differentiation of the determinant $\det M = \det M(U, m_{ud}, m_s, \mu)$ calculated with the reduced matrix formalism of Ref. [22]. The step size in the derivative was chosen small enough to make the systematic error from the finite difference negligible compared to the statistical error. The additive and multiplicative divergences in the condensate were renormalized with the prescription

$$\langle \bar{\psi}\psi \rangle_R(T,\mu) = -\frac{m_{ud}}{f_\pi^4} \left[ \langle \bar{\psi}\psi \rangle_{T,\mu} - \langle \bar{\psi}\psi \rangle_{0,0} \right]. \qquad (13)$$

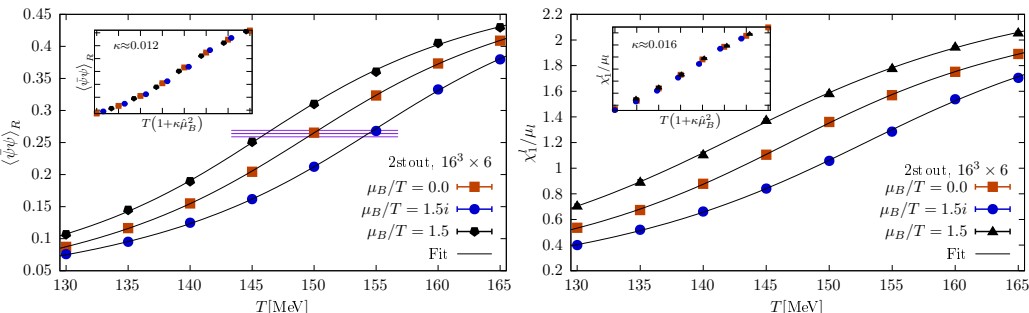

Figure 2: The renormalized chiral condensate (left) and the light quark number-to-light quark chemical potential ratio (right) as a function of $T$ at fixed $\mu_B/T = 1.5, 0$ and $1.5i$ on 2stout mproved lattices at $N_\tau = 6$. The insets show a rescaling of the temperature axis by $T \to T\left(1 + \kappa\left(\frac{\mu_B}{T}\right)^2\right)$, which approximately collapses the curves onto each other if $\kappa \approx 0.012$ and $0.016$ are chosen for the chiral condensate and the quark number-to-chemical potential ratio, respectively.

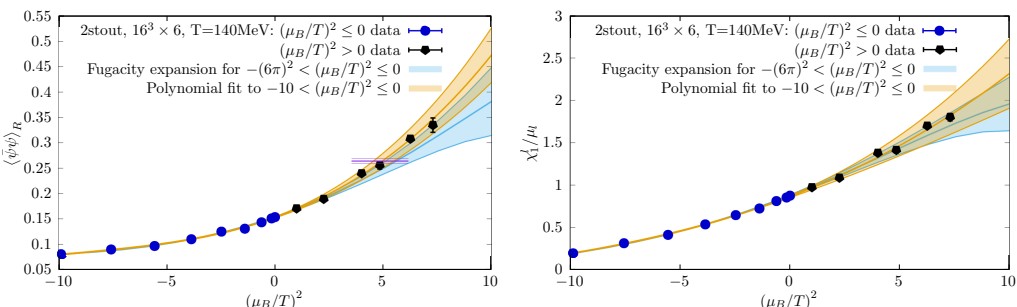

Figure 3: The renormalized chiral condensate (left) and the light quark number-to-light quark chemical potential ratio (right) as a function of $(\mu_B/T)^2$ at temperature $T = 140$ MeV with the 2stout improved staggered action at $N_\tau = 6$. Data from simulations at real $\mu_B$ (black) are compared with analytic continuation from simulations at imaginary $\mu_B$ (blue). In the left panel the value of the condensate at the crossover temperature at $\mu_B = 0$ is also shown by the horizontal line. The simulation data cross this line at $\mu_B/T \approx 2.2$.

We also calculated the light quark density

$$\chi_1^l \equiv \frac{\partial \left(p/T^4\right)}{\partial \left(\mu/T\right)} = \frac{1}{VT^3} \frac{1}{Z(T,\mu)} \frac{\partial Z(T,\mu)}{\partial \hat{\mu}} = \frac{1}{VT^3 \langle \varepsilon \rangle_{T,\mu}^{\mathrm{SQ}}} \left\langle \varepsilon \frac{\partial}{\partial \hat{\mu}} \ln \left| \mathrm{Re} \det M_{ud}^{\frac{1}{2}} \right| \right\rangle_{T,\mu}^{\mathrm{SQ}}. \qquad (14)$$

In this case the derivative on a fixed configuration can be obtained analytically using the reduced matrix formalism. The light quark density does not have to be renormalized.

Our results for a temperature scan between 130 MeV and 165 MeV at real chemical potential $\mu_B/T = 1.5$, zero chemical potential, and imaginary chemical potential $\mu_B/T = 1.5i$ are shown in Fig. 2. We also show that a rescaling of the temperature axis of the form $T \to T\left(1 + \kappa\left(\frac{\mu_B}{T}\right)^2\right)$, where $\kappa \approx 0.012$ for the chiral condensate and $\kappa \approx 0.016$ for $\chi_1^l/\mu_l$ collapses the curves into each other. Such a simple rescaling indicates that up to $\mu_B/T = 1.5$ the chiral crossover does not get narrower, which is what one would expect in the vicinity of a critical endpoint.

Our results for the chemical potential scan at a fixed temperature of $T = 140$ MeV are shown in Fig. 3. We have performed simulations at $\mu_B/T = 1, 1.5, 2, 2.2, 2.5, 2.7$. The sign-quenched results are compared with the results of analytic continuation from imaginary chem-

ical potentials. To demonstrate the magnitude of the systematic errors of such an extrapolation we considered two fits. (*i*) As the simplest ansatz, we fitted the data with a cubic polynomial in $\hat{\mu}_B^2 = \left(\frac{\mu_B}{T}\right)^2$ in the range $\hat{\mu}_B^2 \in [-10, 0]$. (*ii*) As an alternative, we also and ansätze for both $\left\langle \bar{\psi}\psi \right\rangle_R$ and $\chi_1^l/\hat{\mu}_l$ based on the fugacity expansion $p/T^4 = \sum_n A_n \cosh(n\mu_l/T)$, fitting the data in the entire imaginary-potential range $\hat{\mu}_B^2 \in \left[-(6\pi)^2, 0\right]$ using respectively 7 and 6 fitting parameters. Fit results are also shown in Fig. 3; only statistical errors are displayed. While sign reweighting and analytic continuation give compatible results, in the upper half of the $\mu_B$ range the errors from sign reweighting are an order of magnitude smaller. In fact, sign reweighting can penetrate the region $\hat{\mu}_B > 2$ where the extrapolation of many quantities is not yet possible with standard methods [7, 9].

## 4 Conclusions

Due to the increasing computing power of modern hardware, direct approaches to finite density QCD are becoming increasingly feasible, and are opening up a new window to study the bulk thermodynamics of strongly interacting matter from first principles. In this conference contribution and the paper Ref. [1] which it is based on, we studied the method of sign reweighting in detail for the first time. While the method is ultimately bottlenecked by the sign problem, in the region of applicability it offers excellent reliability compared to the dominant methods of Taylor expansion and imaginary chemical potentials - which always provide results having a shadow of a doubt hanging over them due to the analytic continuation problem. We have demonstrated that the strength of the sign problem can be easily estimated with $\mu = 0$ simulations, making the method practical and the planning of simulation projects straightforward. We have also demonstrated that the method extends well into the regime where the established methods start to lose predictive power, and covers the range of the RHIC Beam Energy Scan (BES) [31, 32].

The lattice action used in this study is often the first point of a continuum extrapolation in QCD thermodynamics. Furthermore, while the sign problem is exponential in the physical volume, it is not so in the lattice spacing. Continuum-extrapolated finite $\mu_B$ results in the range of the RHIC BES and is already within reach for the phenomenologically relevant aspect ratio of $LT \approx 3$.

On a more methodological point, the phase and sign reweighting approaches only guarantee the absence of heavy tailed distributions when calculating the ratio of the partition functions (or the pressure difference) of the target and simulated theories. Furthermore, the optimum property of the sign quenched ensemble is only a statement about the denominator of Eq. (1) (right). The optimal ensemble when both the numerator and the denominator are taken into account is most likely, however, observable dependent. For these two reasons, the study of the probability distributions of observables other than the pressure is an important direction for future work.

## Acknowledgements

The project was supported by the BMBF Grant No. 05P18PXFCA. This work was also supported by the Hungarian National Research, Development and Innovation Office, NKFIH grant KKP126769. A.P. is supported by the J. Bolyai Research Scholarship of the Hungarian Academy of Sciences and by the ÚNKP-21-5 New National Excellence Program of the Ministry for Innovation and Technology from the source of the National Research, Development and Innovation Fund. The authors gratefully acknowledge the Gauss Centre for Supercomputing e.V.

(www.gauss-centre.eu) for funding this project by providing computing time on the GCS Supercomputers JUWELS/Booster and JURECA/Booster at FZ-Juelich.

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
