# Peer review of "Lattice simulations of the QCD chiral transition at real $\mu_B$"

_SciPost Physics Proceedings, doi:SciPost Phys. Proc. 6, 002 (2022)_

## Round 1 · Referee Report · Anonymous (Referee 1) · 2022-3-17

Strengths
- Paper is devoted to a very important problem, phase diagram in $\mu_B - T$ plane.
- The authors use the lattice regularization for QCD which provides first principles results.
- The authors propose and use the direct approach to solve the most difficult problem of lattice simulations at nonzero $\mu_B$ - the sign problem.
- The severity of the sign problem is presented in details.
- The results obtained for the transition line in $\mu_B - T$ plane are very encouraging.
Report
The paper is devoted to one of very important problems of the particle physics, the phase diagram in $\mu_B - T$ plane. The understanding of the phase diagram is needed for interpretation of the ongoing and future experiments on heavy ion colliders. The authors study QCD in lattice regularization. It is known that
this approach provides first-principles results. The most severe problem of this approach appears at nonzero $\mu_B$ due to so-called sign problem. The authors propose to use the reweighting approach to solve this problem. This approach is not new but the authors of this paper achieved the most encouraging results.
In the Introduction the authors review the approaches applied so far to the sign problem and results achieved as well as formulate the drawbacks of those approaches. They pay more attention to the reweighting methods and to the overlap problem. In Section II the authors consider the strength of the sign problem for two methods of reweighting. They obtain the estimations for the cost of simulations which is important for practical purposes. The next section is devoted to results. The state of the arts improved gauge field action and improved quark action are used for simulations on moderate size $16^3 \times 6$ lattices. The scans for fixed $T=140$ MeV and for fixed $\mu_B/T=1.5$ were made. Two observables were computed: The renormalized light-quark chiral condensate and the light quark density. The results for the fixed temperature scan are compared with the analytic continuation from imaginary chemical potential. This comparison shows that the reweighting is much more reliable than the analytic continuation in the range of large $\mu_B/T$. I think this result can inspire attempts to improve the analytic continuation approach. The scan for fixed $\mu_B/T$ suggests that the QCD crossover is not changing its width at this value of $\mu_B/T$ in comparison with zero chemical potential.
In my opinion, the paper presents details of the very promising approach to the sign problem solution and provides new interesting results for QCD at finite $\mu_B$. It definitely deserves to be published.
this approach provides first-principles results. The most severe problem of this approach appears at nonzero $\mu_B$ due to so-called sign problem. The authors propose to use the reweighting approach to solve this problem. This approach is not new but the authors of this paper achieved the most encouraging results.
In the Introduction the authors review the approaches applied so far to the sign problem and results achieved as well as formulate the drawbacks of those approaches. They pay more attention to the reweighting methods and to the overlap problem. In Section II the authors consider the strength of the sign problem for two methods of reweighting. They obtain the estimations for the cost of simulations which is important for practical purposes. The next section is devoted to results. The state of the arts improved gauge field action and improved quark action are used for simulations on moderate size $16^3 \times 6$ lattices. The scans for fixed $T=140$ MeV and for fixed $\mu_B/T=1.5$ were made. Two observables were computed: The renormalized light-quark chiral condensate and the light quark density. The results for the fixed temperature scan are compared with the analytic continuation from imaginary chemical potential. This comparison shows that the reweighting is much more reliable than the analytic continuation in the range of large $\mu_B/T$. I think this result can inspire attempts to improve the analytic continuation approach. The scan for fixed $\mu_B/T$ suggests that the QCD crossover is not changing its width at this value of $\mu_B/T$ in comparison with zero chemical potential.
In my opinion, the paper presents details of the very promising approach to the sign problem solution and provides new interesting results for QCD at finite $\mu_B$. It definitely deserves to be published.

---

## Editorial Decision

published